# In situ micropillar compression of an anisotropic metal-organic framework single crystal

Zhixin Zeng [1], Yuan Xiao[2], Jeffrey M. Wheeler [2] & Jin-Chong Tan [1]✉

Understanding of the complex mechanical behavior of metal-organic frameworks (MOF) beyond their elastic limit will allow the design of real-world applications in chemical engineering, optoelectronics, energy conversion apparatus, and sensing devices. Through in situ compression of micropillars, the uniaxial stress-strain curves of a copper paddlewheel MOF (HKUST-1) were determined along two unique crystallographic directions, namely the (100) and (111) facets. We show strongly anisotropic elastic response where the ratio of the Young's moduli are $E_{(111)} \approx 3.6 \times E_{(100)}$, followed by extensive plastic flows. Likewise, the yield strengths are considerably different, in which $Y_{(111)} \approx 2 \times Y_{(100)}$ because of the underlying framework anisotropy. We measure the fracture toughness using micropillar splitting. While in situ tests revealed differential cracking behavior, the resultant toughness values of the two facets are comparable, yielding $K_c \sim 0.5\,\mathrm{MPa}\sqrt{\mathrm{m}}$. This work provides insights of porous framework ductility at the micron scale under compression and failure by bonds breakage.

[1] Multifunctional Materials & Composites (MMC) Laboratory, Department of Engineering Science, University of Oxford, Parks Road, Oxford OX1 3PJ, UK. [2] Laboratory for Nanometallurgy, Department of Materials, ETH Zurich, Vladimir-Prelog-Weg 5, HCI G 503, 8093 Zürich, Switzerland.
✉email: jin-chong.tan@eng.ox.ac.uk

Metal-organic frameworks (MOF) are a versatile family of porous solids constructed from organic and inorganic building blocks, yielding either crystalline or amorphous hybrid materials with vastly tunable physical and chemical properties. MOF crystals and MOF-based composites are being developed to target numerous technological applications, ranging from sensors and dielectrics[1,2], capture and separations[3], electroluminescence and lighting[4,5], to drug delivery[6], mechanical energy absorption[7], and catalysis[8,9]. Although the chemical and sorption-related properties of MOF compounds have been systematically studied in the past 25 years[10], the research on their physical properties, especially pertaining to the mechanical behavior of MOFs, is significantly lacking behind[11–13].

Hitherto, the majority of studies in the field of "MOF mechanics"[14] are focused on the elastic properties of MOF crystals and structural vibrations, employing experimental techniques such as nanoindentation[11], Terahertz and Raman spectroscopy[15,16], and Brillouin scattering[17,18]; while computational studies encompass density functional theory (DFT) calculations of single-crystal elastic constants[19–21] and molecular dynamics of reversible framework deformation[22,23]. Understanding the plastic behavior (beyond the elastic limit)[24] and ultimate fracture[25] of MOFs are central to practical applications for improved mechanical durability and resilience of the resultant devices. To put this into context, compression would be a major loading case for many MOF applications involving powder compaction or consolidation, prevalent in adsorption columns and for manufacture of drug pellets and catalyst pellets, which after plastic deformation required to retain the powder functionalities. In terms of fracture behavior, catastrophic fracture could occur in potential MOF devices such as sensor chips, dielectric films, and photochromic coatings, when subject to thermo-mechanical loading or bending stress in service. Another example concerns energy absorption applications, where cyclic tensile-compressive loading and fatigue crack resistance will be important.

With the advancement of focused ion beam (FIB) milling combined with in situ micromechanical testing, the plastic deformation of a wide range of materials have been reported in the past decade, exemplars of which include superalloys, gallium arsenide, bones, bulk metallic glasses and amorphous silica[26–30]. More recently, a uniaxial micropillar compression study of MOF glass revealed extensive plasticity of the amorphized $a_g$ZIF-62 monolith, at least on the micrometer length scale[31]. However, to the best of our knowledge the precise characterization of the elastic-to-plastic transition (yielding), large-strain deformation (plastic flow), and fracture toughness (cracking) of MOF single crystals by microcompression has not yet been reported.

In this study, $Cu_3(BTC)_2$ (BTC = benzene-1,3,5-tricarboxylate), also known as the "HKUST-1" structure, comprising the copper paddlewheel framework has been chosen as a representative 3-D MOF material to study the stress-strain relationship and fracture behavior of a mechanically anisotropic MOF crystal. HKUST-1 crystallizes in the cubic space group ($Fm\bar{3}m$) and exhibits a large surface area in the range of $1000 \, m^2/g$[32]. The ordered nanoporous structure of HKUST-1, its ease of synthesis, and tunable functionalities due to its open metal sites, made this MOF structure one of the most significant in the field. In contrast to an amorphous MOF glass[31] or the polycrystalline monoliths of HKUST-1[33] that are mechanically isotropic[34], the mechanical response of a single crystal of HKUST-1 is predicted by DFT to be strongly directionally dependent[35]. This gives us the unique opportunity to probe the effects of mechanical anisotropy of a porous MOF, both before and after exceeding the elastic limit and until the point of rupture, by leveraging the in situ micropillar compression method.

## Results and discussion

**Uniaxial compression of micropillars.** In situ micro-compression experiments were conducted on the submillimeter-sized HKUST-1 crystals (*ca.* $200-600 \, \mu m$), which were prepared using a solvothermal method (see Methods)[36]. Activated HKUST-1 crystals were cold-mounted on an epoxy resin (Struers Epofix) with both the (100) and (111) facets exposed (see Fig. 1a) using an established methodology designed for studying MOF single crystals[37]. The sample surface was meticulously prepared to maximize the surface quality, employing glycerol as the non-penetrating polishing liquid. The large crystal surface was rinsed with ethanol and then desolvated at 80 °C to remove residual liquid. Thereafter, the sample was stored in a desiccator until testing (Supplementary Methods and Supplementary Fig. 1).

In order to perform the in situ uniaxial compression test, the HKUST-1 crystals were machined into isolated micropillars using gallium focused ion beam (Ga FIB), see Supplementary Fig. 2. The relatively large diameters of these MOF pillars were chosen to be ~5 μm for the purpose of mitigating the ion damage (all calculations were based on the precise dimensions of the individual pillar, summarized in Supplementary Table 1). Six HKUST-1 micropillars (three for each crystal facet) were uniaxially compressed, at a constant strain rate of $1 \times 10^{-3} \, s^{-1}$. In situ micro-compression tests were performed using the Alemnis instrument equipped with a conical indenter, which has a flat apex in order to exert a uniform compressive force. As shown in Fig. 1b, c, the surface of the samples revealed microscopic pores generated by FIB milling, which is inferred to be only superficial damage rather than larger-scale structural failure on account of the consistency of the load vs. displacement curves measured (*vide infra*).

Under uniaxial compression, the (100) facet of HKUST-1 deformed elastically followed by a visible buckling at the base of the pillar (Fig. 1b). Subsequently, the micropillar sidewall surface experienced micro-buckling and rumpling, which occurred on the layers degraded by the ion beam and this might have contributed to the wider scattering of the force curves for the (100) facet, see Fig. 1d. By contrast, also shown in Fig. 1c, d, the deformation of the (111) facet under the compressive force is considerably more stable. Likewise, there was the elastic deformation before a steady micro-buckling, however, a plastic flow plateau emerged in the stress-strain curves of the (111) facets, in contrast to the apparent non-linear behavior of the (100) facets after an initial average yield point (*Y*) at around 116 MPa.

To examine the stability of the HKUST-1 pillars, we investigated its buckling strength using the Euler's column formula (see Supplementary Note 1). In this column buckling analysis, two possible boundary conditions (BCs) were considered: firstly, the bottom of the pillar was fixed while the top loading surface was free to deflect; secondly, on top of the first scenario, slippage is permitted between the flat punch and the top surface of the compressed pillar. Utilizing the first BC, we determined the Euler's critical load: $P_{E(100)} = 0.85 \pm 0.13 \, mN$ and $P_{E(111)} = 4.51 \pm 1.92 \, mN$. Conversely, employing the second BC, the magnitudes were elevated to $6.96 \pm 1.06 \, mN$ and $36.99 \pm 15.71 \, mN$ for the (100) and (111) facets, respectively. According to the recorded compression test videos (Supplementary Table 2), it can be seen that the compressed (100) micropillars after unloading had sustained some degree of inclination (hence buckled) as opposed to the (111) micropillars that remained upright (without buckling) after unloading. This observation supports the buckling strength hypothesis presented above, where $P_{E(111)} > P_{E(100)}$ independent of the imposed BCs.

**Anisotropic elasticity.** The elastic deformation of both facets was reflected by the linear segment of the loading curves. Accordingly,

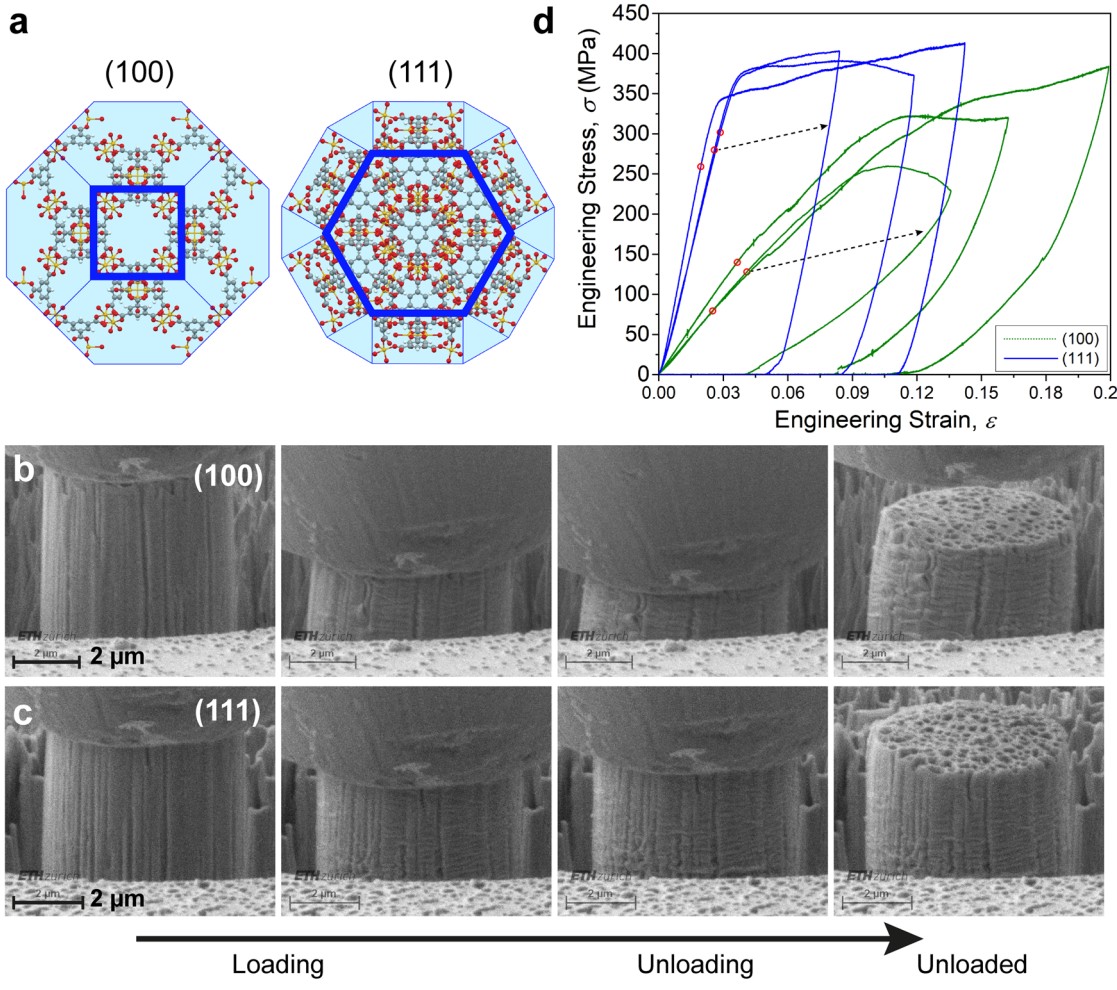

**Fig. 1 Uniaxial compression of HKUST-1 micropillars. a** Porous crystalline framework of the HKUST-1 MOF structure, viewed down the (100) and (111) crystal facets, respectively. Color code: copper (yellow), oxygen (red), carbon (gray), and hydrogen (white). In the periodic structure, the inorganic copper paddlewheel clusters are bridged by the organic BTC linkers. **b, c** Scanning electron microscope (SEM) images of HKUST-1 micropillars milled by focused ion beam and subsequently compressed uniaxially with the axis of the flat punch oriented normal to the (**b**) (100)- and (**c**) (111)-crystal facets, respectively. Recorded videos of the in situ compression tests are given in Supplementary Table 2. **d** Engineering stress-strain curves obtained from the compression tests on the (100) and (111) crystal facets, showing their distinctively different elastic-plastic deformation behavior. The yield points (red circles) were established using the first-derivative threshold method (see Supplementary Fig. 4 and Supplementary Note 2).

we have measured the Young's moduli ($E$) of the two HKUST-1 facets, employing uniaxial compression: $E_{(100)} = 3.44 \pm 0.42$ GPa and $E_{(111)} = 12.37 \pm 1.79$ GPa. Notably, these stiffness values are in good agreement with the reported theoretical predictions by density functional theory (DFT) calculations of the single-crystal elastic constants of HKUST-1 ($E_{(100)} \approx 3$ GPa and $E_{(111)} \approx 15$ GPa)[35]. Furthermore, we have used the instrumented nanoindentation technique (IIT) to measure the stiffness anisotropy, employing the MTS Nanoindenter XP instrument equipped with a Berkovich indenter tip (Supplementary Fig. 3a). In comparison with the results obtained using IIT where the indentation moduli ($M$ used in place of $E$ when the sample's Poisson's ratio, $\nu_s$ is unknown)[38] were determined as: $M_{(100)} = 7.4 \pm 0.6$ GPa and $M_{(111)} = 10.4 \pm 0.8$ GPa (see Supplementary Fig. 3b). The uniaxial compression test has excellent agreement with theory, overcoming the averaging effect that was previously affecting the accuracy of IIT when measuring strongly anisotropic crystals. This is a remarkable result, because *via* uniaxial micropillar compression, we have directly measured the anisotropic Young's moduli of HKUST-1 crystal at the highest accuracy thus far attainable. This finding further supports the notion that the influence of the damaged top surface on the modulus measurement was superficial or even negligible, and there

was no severe amorphization of the HKUST-1 structure caused by FIB milling.

**Yield strength and plastic flow**. As shown in Fig. 1d, both of the (100) and (111) facets in HKUST-1 develop significant work hardening, to the extent that there is no well-defined yield strength ($Y$). We took the yield stress where it turned into a non-linear portion as the yield strength, *viz.* the proportional limit, even so the yield point may be marginally offset. Despite the linear elastic part of the stress vs. strain ($\sigma - \varepsilon$) curves for the HKUST-1's (111) facet is comparably more discernible, to determine the yield point in a systematically reproducible way, we calculated the first derivative of the stress-strain data ($d\sigma/d\varepsilon$) and then utilized the moving average approach to smooth out the first-derivative curve to highlight the fluctuations and trends (Supplementary Fig. 4). On this basis, we have determined the yield strength of the (111) crystal facet, which is $Y_{(111)} = 280.2 \pm 21.4$ MPa. Whereas the stress-strain curves obtained from compression test of the (100) facet are not as consistent, resulting in $Y_{(100)} = 115.7 \pm 24.1$ MPa. Hence we observed that $Y_{(111)} \approx 2Y_{(100)}$, which can be rationalized by considering the

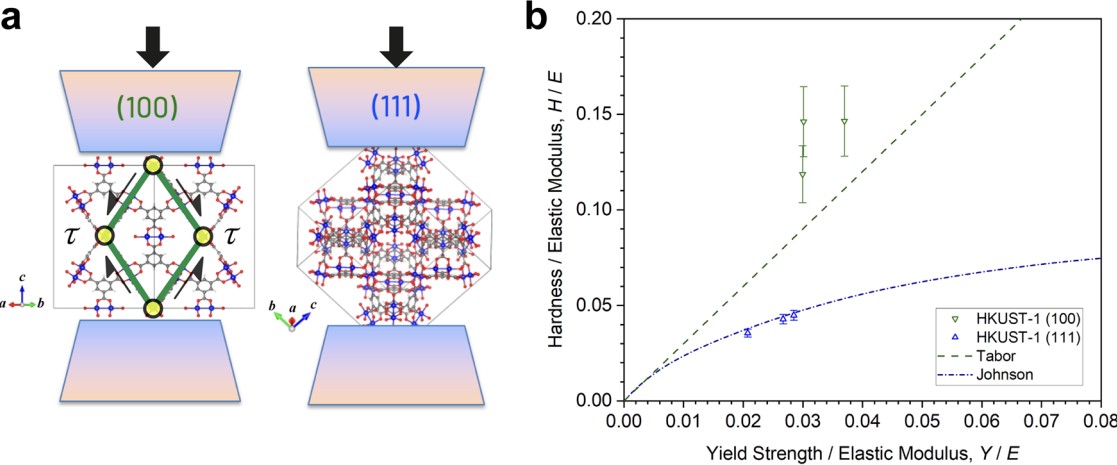

**Fig. 2 Mechanical anisotropy of (100) and (111) crystal facets. a** Schematics (not to scale) showing the plausible source of mechanical anisotropy associated with the directional effects in the HKUST-1 framework when compressing the framework structure on the (100)- and (111)-crystal facets. Orientations susceptible to shear-induced framework distortions are designated as $\tau$ by two pairs of shear stresses. Copper paddlewheel SBUs are denoted by yellow nodes (joints), while BTC linkers in green. Color scheme for atoms: copper in blue, oxygen in red, carbon in grey, hydrogen in white. **b** $H/E$ vs. $Y/E$ for the two HKUST-1 facets in comparison with the Tabor and Johnson correlations. The error bars represent the standard deviation.

anisotropic HKUST-1 framework when loaded along the two very dissimilar crystallographic directions, as illustrated in Fig. 2a. We reasoned that the bond density is much higher for (111) with less shear applied to the bonds and secondary building units (SBU = copper paddlewheels), while the bonds for (100) are oriented parallel to the shear suggesting buckling would occur. In this context, the SBUs serve as 'flexible' nodes to facilitate the angular distortion of the parallelepiped-like structure, comprising 'rigid' BTC ligands as illustrated in Fig. 2a. This proposed mechanism is also consistent with previous DFT prediction that the shear modulus of the HKUST-1 structure is minimum in the inclined direction ($G_{\langle 110 \rangle} \sim 1$ GPa)[15].

Subsequently, we analyzed the energy absorbed and released in the uniaxial compression test during elastic deformation. The calculated modulus of resilience (i.e., integrated area under $\sigma-\varepsilon$ curve $\leq Y$) of the HKUST-1's (100) facet equals to $U_{R(100)} = 2.63 \pm 0.03$ J/m$^3$, while it is $U_{R(111)} = 3.33 \pm 0.84$ J/m$^3$ for the (111) facet. Therefore, subject to compression on the (111) crystallographic facet, a unit volume of the HKUST-1 crystal absorbs ~26.6% more strain energy than that of the (100) facet without resulting in plastic deformation.

Unlike the elastic modulus and yield strength values obtained using the micropillar compression test, the values of the hardness ($H$) for the two facets measured by the IIT experiments are quite similar: $H_{(100)} = 463 \pm 58$ MPa and $H_{(111)} = 491 \pm 28$ MPa (see Supplementary Fig. 3c). During indentation, the material around the indenter is displaced by plastic flow and contributes to its elastic and plastic response. To gain a better understanding of the structural response of the HKUST-1 framework, we studied the correlation between the hardness ($H$) and yield strength ($Y$), when normalized by its Young's modulus ($E$). As can be seen in Fig. 2b, the $H/E$ ratio is often correlated with the $Y/E$ ratio when the latter is sufficiently large[39]. We found that the $H/E$ vs. $Y/E$ relation for HKUST-1 (111) is consistent with the result depicted by the cavity theory proposed by Johnson[40]. In this theory, the discrepancy of the volume displaced by the indenter and the elastic expansion is explained by material movement from the plastic deformation zone into the cavity, which is plausible considering the nanoporous nature of MOFs, where framework collapse is expected[41,42]. For HKUST-1, the alignment of the largest pores normal to the (111) facet may expedite cavity collapse and volume change associated with plastic deformation.

In general, for different materials, the hardness increases as the yield strength grows due to the reduction of the elastic deformation, which contributes to a smaller contact area. As shown in Fig. 2b, the $H/E$ ratio of the (100) facet is approximately 3.4 times greater than that of the (111) facet. The strength of the framework for the (100)- and (111)-crystal facets should differ considerably (Fig. 2a), which is also reflected by the measured Young's moduli, where $E_{(111)} \approx 3.6 \times E_{(100)}$. Tabor stated that the hardness of rigid plastic materials[43] and metals with work hardening[44] is typically about three times their yield strength ($H \approx 3Y$). However, as established in Fig. 2b, the (100) facet of HKUST-1 exceeds the three-fold relation, which could be ascribed to two possible factors: (i) in Tabor's study, the three-fold correction was used to approximate the mean contact pressure under a spherical indenter for a dense material, but this approximation is not applicable to our case as HKUST-1 has a collapsible nanoporous structure. (ii) As evidenced in Fig. 1b (and Supplementary Movies P1−P3), the (100)-orientation is more susceptible to buckling in its unconfined state, causing its yield strength to be low, while its confined state (during hardness testing) does not permit the framework to buckle following its natural tendency. Consequently, its hardness is even higher than it would be naturally due to its yield strength.

**Fracture toughness measured by micropillar splitting**. For investigating the anisotropic mechanical behavior of HKUST-1 beyond elasticity towards the complete structural failure, a fracture mechanics-based micropillar splitting test was performed on the (100)- and (111)-crystal facets, see Fig. 3a, b. A cube-corner indenter was used to apply the splitting stress starting from the centers of the pillars for both crystal orientations. The abrupt drop in the load in Fig. 3c corresponds to the rapid and unstable crack propagation. The critical splitting load is indenter geometry-dependent. For instance, the magnitude of the splitting load will be higher if an indenter tip such as a Berkovich tip of larger included angle is employed than that of the sharper cube-corner indenter tip, because of higher stress intensity of the latter. Using the load vs. displacement curves of Fig. 3c, the work done by the splitting load was calculated: $W_{(100)} = 238.2 \pm 45.4$ J m$^{-2}$; $W_{(111)} = 51.2 \pm 10.7$ J m$^{-2}$.

Subsequently, we applied the reported outcome of cohesive-zone finite element method[45] to determine the relationship

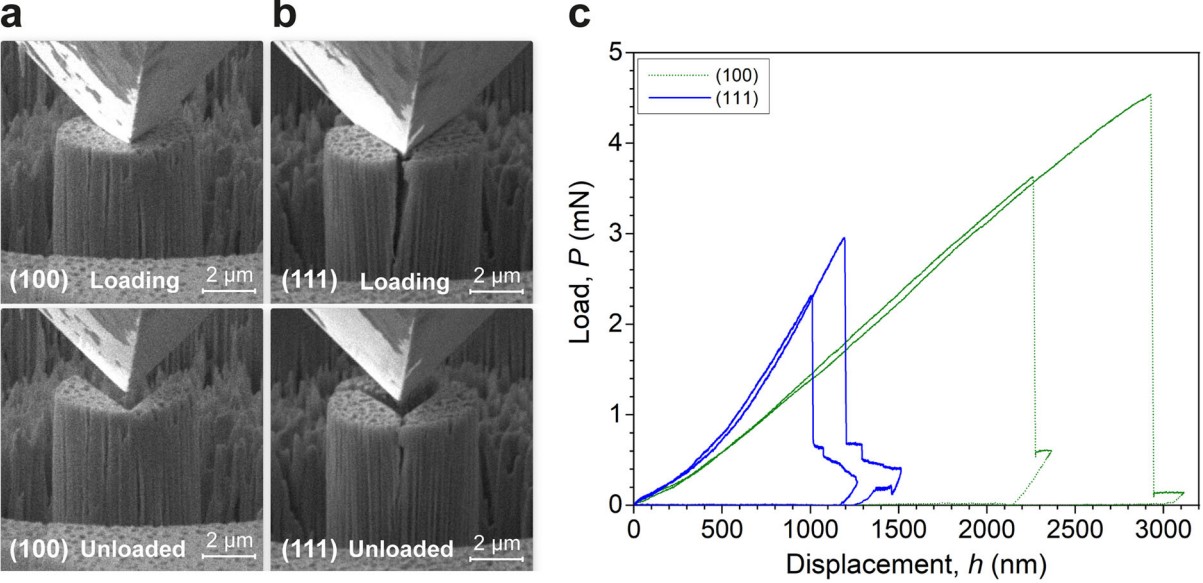

**Fig. 3 In situ splitting of (100)- and (111)-oriented micropillars.** In situ SEM images of the HKUST-1 micropillars in the splitting tests, when compressed on the (**a**) (100)- and (**b**) (111)-crystal facets. Recorded videos of the in situ tests are given in Supplementary Table 2. **c** Load vs. displacement (*P-h*) curves acquired from micropillar splitting of the two crystal facets.

between the stress intensity, the critical load, and the *E/H* ratio for the two facets. The fracture toughness ($K_c$) of the crystals is expressed by[46]:

$$K_c = \gamma \frac{P_c}{R^{3/2}} \qquad (1)$$

where $P_c$ represents the critical load at splitting; $R$ denotes the radius of the pillar; and $\gamma$ is the dimensionless coefficient determining the position of the instability within the pillar, which is associated with the *E/H* ratio. $\gamma$ is also temperature-dependent and material-specific elastic-plastic property[47]. Eq. (1) is widely-used for calculating fracture toughness, because the knowledge of crack dimension and sample geometry is not essential. This is a significant advantage for nanomaterials, especially MOFs; where the crack dimension can be difficult to measure, and the propagation of cracks is usually less stable. It is worth noting that a material under a non-plane strain condition usually experiences larger-scale plastic deformation, and this means that $K_c$ is not independent of the size of cracks or defects in the HKUST-1 crystals and the geometry of the specimen. Consequently, the magnitude of the fracture toughness measured here can be considered only as the upper-bound compared with possible other techniques such as single-edge pre-cracked beam method, reported for an isotropic MOF glass of ZIF-62[48]. Herein, we applied the relationship between $\gamma$ and *E/H* (Supplementary Fig. 5) to determine the averaged values of $\gamma$ for both the (100) and (111) facets: 0.543 and 0.828, respectively. As a result, we calculated the fracture toughness: $K_{c(100)} = 0.524 \pm 0.033\,\text{MPa}\sqrt{\text{m}}$, which is resembling the magnitude of the other facet, $K_{c(111)} = 0.515 \pm 0.066\,\text{MPa}\sqrt{\text{m}}$ in spite of their rather different behavior in terms of the *E/H* ratio and the load vs. displacement curves.

HKUST-1 crystals are predicted to be highly anisotropic from previous DFT calculations (Zener anisotropy ~5.41 using the B3LYP functional)[35]. The similarity of the measured fracture toughness of the two facets is reasoned by the fact that the pillar splitting method imposes three-fold symmetry upon the crack propagation, which averaged out the effect of structural anisotropy. Furthermore, the measured critical stress intensities have been shown to be largely independent of displacement rate

effects, within the range used in these measurements[47]. Unlike the compression test, the damage induced by the FIB milling may not be negligible here since the splitting test of silicon pillars was reported to significantly increase the apparent fracture toughness at small pillar sizes, although the influence diminishes to negligibility at the pillar diameters larger than 10 μm[47]. Therefore, further experiments will be required to investigate the influence of the pillar diameter on the fracture toughness of HKUST-1 pillars, although a convergence of the obtained toughness value is expected to be observed.

**Comparison of fracture toughness values.** To consider these results in a broader material context, the fracture toughness of the single crystal of HKUST-1 can be presented in an Ashby-style plot of $K_c$ versus $E$, as presented in Fig. 4. Thus far, there are only a limited number of studies which have characterized the fracture toughness of MOFs and inorganic-organic framework materials. Their values are by and large clustered near to the projected lower-bound for $K_c$ while occupying a gap unpopulated by conventional materials. Of note, the toughness of HKUST-1 crystal is remarkably higher compared with the amorphous ZIF-62 glass ($K_c = 0.104\,\text{MPa}\sqrt{\text{m}}$)[48], as well as against the nanostructured poly-crystalline monoliths of ZIF-8 ($K_c = 0.074\,\text{MPa}\sqrt{\text{m}}$) and ZIF-71 ($K_c = 0.145\,\text{MPa}\sqrt{\text{m}}$)[25]. As these values were obtained using different geometries and techniques, the comparison is not direct, but the scale of the difference implies that significantly higher toughness is likely in the HKUST-1 MOF. For direct comparison to engineering materials in Fig. 4, however, this will require the production and testing of bulk crystals (cm scale) to standard ASTM fracture geometries, which are currently intractable in the MOF field.

In summary, we have conducted the uniaxial micro-compression, instrumented nanoindentation, and micropillar splitting tests on large (sub-mm) single crystals of the copper paddlewheel MOF, HKUST-1, oriented on the (100) and (111) facets. Based on the in situ recorded stress-strain curves and fracture history, we have established the anisotropic elastic moduli, yield strength, plastic flow, and fracture toughness of the porous HKUST-1 structure in two distinct crystallographic directions. Moreover,

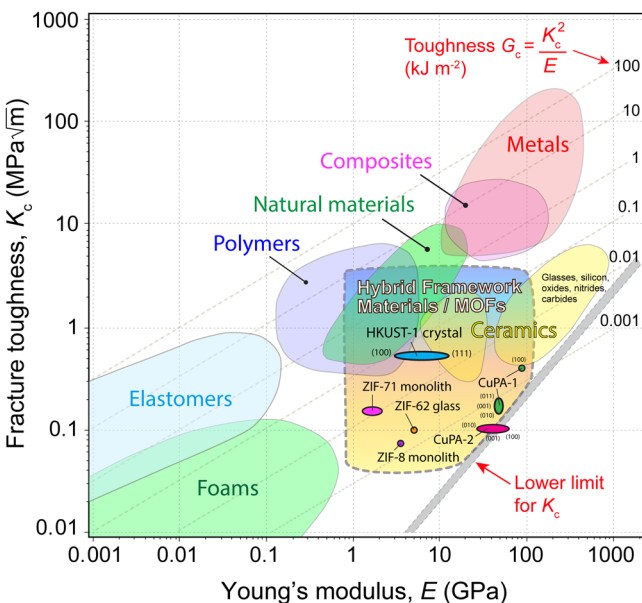

**Fig. 4 Fracture toughness materials selection map incorporating MOF structures.** Ashby-style plot of fracture toughness ($K_c$) versus Young's modulus ($E$) of HKUST-1 single crystal against common engineering materials and hybrid framework materials, including dense inorganic-organic frameworks[38], nanocrystalline MOF monoliths (ZIF-8 and ZIF-71)[25], and amorphous ZIF-62 glass[48]. The fracture toughness plotted for HKUST-1 crystal represents the upper-bound value using the micropillar splitting technique. Note this is an indirect comparison due to the different sample geometries and test techniques employed.

the experimental results of the anisotropic Young's moduli obtained by uniaxial compression demonstrate the veracity of the theoretical values from previous DFT calculations. Insights gained from the plastic flow under uniaxial compression (a sign of ductility) and fracture behavior of HKUST-1 should instigate exciting future research to determine the underpinning mechanisms; several avenues are outlined below alongside foreseeable challenges. High-resolution imaging may be pursued by cryo-TEM although the electron beam sensitivity of MOFs will likely be a barrier for in situ micro-compression tests under TEM. The viscoelastic/plastic response and stress relaxation mechanism of MOF materials can be investigated via both variable monotonic strain rate and strain-rate jump micro-compression testing. It will be key to systematically measure fracture toughness by cyclic microcantilever bending for J-integral evaluation of MOF-type materials to validate the more rapid FIB micropillar splitting and nanoindentation fracture approaches, to obtain an improved understanding of the relative toughness values of MOF crystals (porous vs. dense) and monoliths, extending to inorganic-organic glasses, covalent organic frameworks (COFs) and beyond.

## Methods

**Synthesis of large HKUST-1 single crystals**. The solvothermal synthesis of HKUST-1 (Cu$_3$BTC$_2$, BTC = benzene-1,3,5,-tricarboxylate) uses glacial acetic acid as a modulator since it is effective and reproducible to yield "large" (sub-mm) crystals of about 200−600 μm (Supplementary Fig. 1). Firstly, 0.49 g of Cu(NO$_3$)$_2$·3H$_2$O was fully dissolved in 3 mL of deionized water before combined with 3 mL of $N$, $N'$-dimethylformamide (DMF) in a scintillation vial (20 mL), forming the Cu(NO$_3$)$_2$ solution. Meanwhile, we dissolved 0.24 g of trimesic acid (H$_3$BTC) in 3 mL of ethanol (mild heating to assist with dissolution). Thereafter, the H$_3$BTC solution and 12 mL of glacial acetic acid were successively added to the Cu(NO$_3$)$_2$ solution. The vial containing the mixture of solutions were sealed and heated in oven at 55 °C for 3 days, the reaction yielded blue crystals of HKUST-1 forming on both the wall and the bottom of the vial. The mother liquor was then removed, and the crystals were immersed in ethanol for at least 3 days for solvent exchange. It is worth noting that crystals collected from the bottom of the vial were

susceptible to growth defects, since they were prone to form defects compared with crystals attached on the wall of the glass vial. And thus, it is advisable to store the crystals harvested from the wall and the bottom separately in different vials of ethanol. If needed, the crystals can later be activated in a vacuum oven at 120 °C overnight. The activated crystals are desolvated and free from axially-bound water, evidenced from reversible color change between light and deep blue[49].

**Micropillar compression and micropillar splitting tests**. Sample micro-geometries were produced using a Helios 600i (Thermo Fisher Scientific) FIB workstation operated at an accelerating voltage of 30 keV using a Ga$^+$ focused ion beam. For in situ tests, pillars were fabricated with a target diameter of ~5 μm and an aspect ratio of ~3. A two-step milling method was employed with milling currents of 0.79 nA for coarse milling and 40 pA for fine polishing. Micropillar compression testing was performed using an in situ nanomechanical testing system (Alemnis AG, Thun, Switzerland) inside a Vega 3 (Tescan, Brno, Czech Republic) SEM. An electrically-conductive diamond flat punch tip with an 8 μm diameter was used to optimize imaging quality. Micropillars were compressed at a constant displacement rate appropriate to generate a strain rate of $1 \times 10^{-3}$ s$^{-1}$; this is a common strain rate for microcompression testing as the test takes ~5 min per pillar. The alignment of the flat punch with the FIB crystal facet is typically ± 0.2°. Engineering stress-strain curves were determined from load-displacement curves, using the top diameter of the pillars after correcting for pillar sink-in and instrument compliance[50]. Pillar splitting tests were carried out using the same testing system as used for the micropillar compression tests. A diamond cube corner indenter was used to perform the splitting at a speed of 15 nm s$^{-1}$ under displacement control.

**Instrumented nanoindentation tests**. Nanoindentation experiments were performed using an MTS Nanoindenter XP instrument equipped with a Berkovich indenter tip. Polished HKUST-1 crystals mounted on epoxy resins were indented by continuous stiffness measurement (CSM) method, at a constant strain rate of $5 \times 10^{-2}$ s$^{-1}$, to a maximum surface penetration depth of 2000 nm. Thermal drift measurements were performed at 90% unload.

## Data availability
The primary data that support the findings of this study are available within the supplementary information files and from the corresponding author upon request.

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

## Acknowledgements

This research was funded in whole, or in part, by the UKRI (EP/R511742/1). For the purpose of Open Access, the author has applied a CC BY public copyright license to any Author Accepted Manuscript version arising from this submission. J.C.T. and Z.X.Z. thank the ERC consolidator Grant (PROMOFS grant agreement 771575) for additional funding. The authors would like to thank C. Zaubitzer (ScopeM, ETH Zürich) for assistance with the FIB of the samples. We are grateful to Profs. Steve Roberts and David Armstrong for access to the nanoindentation facility at Oxford Materials.

## Author contributions

J.C.T. and J.M.W. conceived the project. Z.X.Z. prepared the crystal samples, performed data analysis, and written first draft of the manuscript under the supervision of J.C.T.; Y.X. performed the FIB of micropillars and conducted in situ compression experiments under the supervision of J.M.W. All authors discussed the results and contributed to the final version of the manuscript.

## Competing interests

The authors declare no competing interests.
