## [Peer Review File · Communications Chemistry]

Reviewers' comments:

Reviewer #1 (Remarks to the Author):

The manuscript submitted by Zhixin et al. is interesting to read. To the best of my knowledge, such in-situ micropillar compression on MOFs is not reported previously, therefore, this work carries significant novelty in terms of experimental approach. The senior authors are established researchers and known well in their respective areas of research. The article is well-written, and results are clearly explained. Overall, this manuscript can be accepted after a minor revision.

The following are my comments for authors to consider while revising the article.

1. Why was the interest behind reaching strain rate $1\text{E-}3\text{ s}^{-1}$?
2. P-h curves in FigS3 shows pop-outs. Are there phase transitions happening? or what is the reason for such special event?
3. How was the surface roughness taken care while quantifying the mechanical properties?
4. What is the plastic deformation mechanism observed in MOFs through this micropillar compression experiment [w.r.t to the underlying crystal structure]?
5. I am not convinced with the comparison of fracture toughness measurements done on micropillars with the experiments done on bulk crystals.
6. Why was the pillar diameter chosen to be $3\text{ }\mu\text{m}$? It would be interesting to see the deformation behavior of less than a micron diameter pillars with less pore volume.

Reviewer #2 (Remarks to the Author):

In this manuscript, in situ micropillar compression experiments are performed on the MOF-system HKUST-1. Anisotropic effects in the elastic and plastic response are analyzed in detail. Such studies on MOF crystals are rare and a good overview of the existing literature is, to the best of my knowledge, presented. The study is well done, manuscript is well written, and results are interesting and relevant for both the MOF and mechanical properties communities.

In my opinion, the manuscript should therefore be published following minor revision (see suggestions below). My only major concern is the lack of new chemistry in this paper, that is, the current version of the manuscript might be a better fit in, e.g., "Communications Materials". To address this, some more discussion on the chemical/structural origin of the differences in mechanical response among the (100) and (111) facets would be recommended and interesting to include (see also comment #4 below).

1- The study could have been motivated in more details in the Introduction. The authors mention that plastic and fracture behavior are important for the "practical applications" of MOFs, but these applications range from drug delivery to mechanical energy absorption. The main failure mechanism of MOFs in such applications would likely differ, which would affect what type of mechanical characterization would be most relevant.

2- Micropillar samples are made by using focused ion beams, and the authors noted that "the surface of the samples revealed microscopic pores generated by FIB milling". Besides the reported agreement with previous DFT simulations, what evidence do the authors have that the milling indeed only induces "superficial damage" that does not impact the findings? In relation to this, have the authors checked for any structural changes upon milling (partial disordering for example)?

3- Some discussion of size and strain rate effects for HKUST-1 would be interesting to include. For many materials, it is well known that for example the extent of plasticity depends on the specimen size and deformation rate.

4- The fracture toughness of the HKUST-1 system is reported to be higher than that of other MOF materials. Could the authors comment on the structural origin of this observation?

5- The nanoindentation experiments are not explained in the Methods section.

6- Methods section: Please check "(Supplementary Error! Reference source not found.)"

Reviewer #3 (Remarks to the Author):

This paper describes, in detail, the elastic limit in the very well-known MOF HKUST-1 along two crystallographic directions. There is a paucity of work detailing the mechanical properties of MOFs, something essential to their use in real-world applications. The experimental work carried out by Tan et al., is to an incredibly high standard, using state of the art techniques in an area that is ripe for exploration. This paper would be well received by the community as a consequence.

I have only a few minor corrections, and suggestions.

Micropillars are compressed along perpendicular to the (100) and (111) directions. For (100), this would just be along a unit cell axis-direction, for the (111), along the body diagonal of the cube. How were the authors assured that these were the directions in the experiment? This applies to all the measurements, including fracture toughness. Notably the authors flip from (100) to [100], in this cubic structure, this would equate to the same thing, perhaps stick with [100] throughout?

Page 3 In 69, how confident are you that the crystals are 'activated'? And in what sense? Is this solvent loss as well as removal of the bound axial water molecule to reveal an open metal site, and if so what is the evidence for this? If this could be clarified it would be very much appreciated.

Page 7, In 145. You state that 'severe amorphization of the HKUST-1 structure caused by FIB milling' is there a way to quantify this? Is anything known about amorphous HKUST-1 at all?

Page 8, Figure 2. I really like this Figure, however can anything more be said, chemically about the difference observed here between the two directions? What on the surface of the crystal on these surfaces, are they different? There are three independent pore structures in HKUST-1, do you see alignment with the largest pore along the 111 giving rise to the difference? I guess what I'm asking, is whether you compress/fracture the SBU ore along one direction than the other?

Page 14 Spotted 'Supplementary Error! Reference source not found.)' needs fixed.

Response to Reviewers:

Reviewer #1:

The manuscript submitted by Zhixin et al. is interesting to read. To the best of my knowledge, such in-situ micropillar compression on MOFs is not reported previously, therefore, this work carries significant novelty in terms of experimental approach. The senior authors are established researchers and known well in their respective areas of research. The article is well-written, and results are clearly explained. Overall, this manuscript can be accepted after a minor revision.

The following are my comments for authors to consider while revising the article.

1. Why was the interest behind reaching strain rate $1\text{E-}3\text{ s}^{-1}$?

Response: This is a very common strain rate for microcompression testing, as the test takes approximately 5 mins per pillar. We have included this information to Page 16 of the revised manuscript.

2. P-h curves in FigS3 shows pop-outs. Are there phase transitions happening? or what is the reason for such special event?

Response: These are not pop-outs. In Fig.S3(a) of the SI, the steps observed at ca. 5 mN (90% unload) show the displacement drift measured during a constant load hold period for thermal drift checks in nanoindenter. This is a standard procedure for characterising thermal drifts. A note has been added to caption of Fig.S3 in the revised SI to explain this effect. The same reasoning applies to Fig.3 in the main manuscript, albeit at a lower load. Here, one of the (111)-oriented pillars shows some displacement drift during a constant load hold period for thermal drift checks, but this was caused by the fractured pillar sliding against the tip.

3. How was the surface roughness taken care while quantifying the mechanical properties?

Response: Surface roughness was neglected, since it appeared to have an insignificant effect on the contact during microcompression – no non-linearity during initial loading phase of Fig.1(d).

4. What is the plastic deformation mechanism observed in MOFs through this micropillar compression experiment [w.r.t to the underlying crystal structure]

Response: The exact mechanism is not yet understood. The large-strain micropillar experiments did reveal significant plastic flow is achievable under compressive loading at microscopic scale, with very different yielding response between the (100) vs. (111) facets (Fig.1(d)), thus indicating a strong correlation to the underlying crystal structure. From Figures 1(a) and 2(a), we can reason that the bond density is much higher for (111) with less shear applied to the bonds and secondary building units (SBU = copper paddlewheel clusters), while the bonds for (100) are oriented parallel to the shear suggesting buckling would occur. This proposed mechanism is also consistent with previous DFT prediction that shear modulus is minimum in inclined direction ($G_{\langle 110 \rangle} \sim 1 \text{ GPa}$) [ref.14]. Subsequent to shear-induced failure, pore collapse and densification of the framework are expected, being facilitated by breakage of chemical bonds and potentially some bond switching.

Given the strong directional nature of the coordination and covalent bonding of MOFs, the dislocation mechanism prevalent in metals (non-directional bonding) is not expected here in the case of HKUST-1 framework as the energy involved would be unfeasibly large. Confirmation of the precise mechanism, however, will require further studies, possibly by means of cryo-HRTEM to image the plastically deformed structures (and ideally deformed in situ within a TEM holder); even this technique will present further major challenges because of high electron beam sensitivity of MOF materials.

A description on the proposed mechanisms and future opportunities to interrogate the plastic mechanisms have been added to the revised manuscript (Pages 8-9 & 15).

5. I am not convinced with the comparison of fracture toughness measurements done on micropillars with the experiments done on bulk crystals.

Response: This is a valid concern. The micropillar splitting technique has been successfully validated for a range of ceramic materials in the literature with similar toughness values (DOI: 10.1016/j.matdes.2019.107762). This may not hold for MOFs, but this is a first measurement of this type. At present, bulk monolithic MOFs cannot yet be fabricated to the size required for standard ASTM fracture geometries (cm scale), so this comparison is

challenging to make. We have added a note on Page 13 of the revised manuscript (also in caption of Figure 4) to emphasise this outstanding challenge in MOF research.

6. Why was the pillar diameter chosen to be 3 μm ?. It would be interesting to see the deformation behavior of less than a micron diameter pillars with less pore volume.

Response: From the initial FIB milling tests, it was determined that submicron pillars would not mill well into a regular pillar geometry. 3 micron diameter was chosen to present a size less affected by the damage and more reliable for mechanical measurements. Smaller pillars may be achieved using cryo-FIB techniques with very low accelerating voltages, but this was chosen to be the optimal size for this initial study.

Reviewer #2:

In this manuscript, in situ micropillar compression experiments are performed on the MOF-system HKUST-1. Anisotropic effects in the elastic and plastic response are analyzed in detail. Such studies on MOF crystals are rare and a good overview of the existing literature is, to the best of my knowledge, presented. The study is well done, manuscript is well written, and results are interesting and relevant for both the MOF and mechanical properties communities.

In my opinion, the manuscript should therefore be published following minor revision (see suggestions below). My only major concern is the lack of new chemistry in this paper, that is, the current version of the manuscript might be a better fit in, e.g., "Communications Materials". To address this, some more discussion on the chemical/structural origin of the differences in mechanical response among the (100) and (111) facets would be recommended and interesting to include (see also comment #4 below).

Response: Thanks for the helpful suggestions. In the revised manuscript, we have added discussion on the differential mechanical response of the (100) and (111) facets where possible, or we point out to readers the plausible mechanisms that will need further research.

1- The study could have been motivated in more details in the Introduction. The authors mention that plastic and fracture behavior are important for the "practical applications" of MOFs, but these applications range from drug delivery to mechanical energy absorption. The main failure mechanism of MOFs in such applications would likely differ, which would affect what type of mechanical characterization would be most relevant.

Response: In the revised manuscript, we have enhanced the Introduction (Page 2) to better illustrate which mechanical characterisation will be relevant to the type of intended application. In essence, compression would be a major loading case for many MOF applications involving powder compaction or consolidation, experienced by adsorption columns and in the manufacture of drug pellets and catalyst pellets expected to function optimally after plastic deformation. In terms of fracture behaviour, due to MOF brittleness catastrophic fracture could occur in potential devices such as sensor chips, dielectric films,

and photochromic coatings, when subject to thermo-mechanical stresses during service. For energy absorption applications, cyclic tensile-compressive loading and fatigue resistance will be relevant.

2-Micropillar samples are made by using focused ion beams, and the authors noted that “the surface of the samples revealed microscopic pores generated by FIB milling”. Besides the reported agreement with previous DFT simulations, what evidence do the authors have that the milling indeed only induces “superficial damage” that does not impact the findings? In relation to this, have the authors checked for any structural changes upon milling (partial disordering for example)?

Response: Further to the agreement of the anisotropic Young’s moduli (E) with DFT predictions, the measured yield strengths also show good reproducibility where $Y_{(111)} = 280.2 \pm 21.4$ MPa and $Y_{(100)} = 115.7 \pm 24.1$ MPa, supporting the argument that the FIB-induced damage is superficial. Direct X-ray evidence is not available, however, because any structural characterization afterwards would involve more FIB sectioning that will induce further damage to the micropillar.

3-Some discussion of size and strain rate effects for HKUST-1 would be interesting to include. For many materials, it is well known that for example the extent of plasticity depends on the specimen size and deformation rate.

Response: About size effects, we do not expect this to be relevant to MOFs as the deformation mechanism doesn’t have any dislocation basis (see also response to Reviewer 1’s Q4). Rate effects are surely present, but this was outside the scope of this initial investigation. Some discussion on these points have been added to Page 15.

4-The fracture toughness of the HKUST-1 system is reported to be higher than that of other MOF materials. Could the authors comment on the structural origin of this observation?

Response: There are limited data at present on MOF toughness to establish any credible structure-property relationships. So Figure 4 basically is to show the ‘bubble’ in which MOF materials are projected to lie in terms of K_{IC} values, but the structural origin is not yet clear.

Now it is broadly understood that for polycrystalline monoliths of ZIF-8 and ZIF-71, measured by cube-corner nanoindentation, their toughness is dominated by grain boundary sliding mechanism of constituent nanosized crystals (ref.24). For ZIF-62 glass, which is amorphous and measured by bending of precracked beams, its toughness is attributed to pre-existing surface flaws and breakage of Zn-N bonds in crack propagation (ref.47). Since this is a first study of MOF fracture toughness by micropillar splitting of HKUST-1 crystals, the K_{IC} values derived from other geometries and techniques may not be directly comparable (a note has been added to caption of Figure 4). Above notwithstanding, we may propose for HKUST-1 crystal the higher toughness might be linked to the low density of surface flaws, differential surface energies of (100) vs (111) facets, absence of grain boundaries (i.e. single crystal), and energy dissipation by compressibility of nanopores. The precise mechanism will need further investigations, see additions to Page 15.

5- The nanoindentation experiments are not explained in the Methods section.

Response: A description has been added to the Methods (Page 17) on instrumented nanoindentation testing.

6- Methods section: Please check “(Supplementary Error! Reference source not found.)” **Response:** We have corrected the error as “Supplementary Figure S1”.

Reviewer #3:

This paper describes, in detail, the elastic limit in the very well-known MOF HKUST-1 along two crystallographic directions. There is a paucity of work detailing the mechanical properties of MOFs, something essential to their use in real-world applications. The experimental work carried out by Tan et al., is to an incredibly high standard, using state of the art techniques in an area that is ripe for exploration. This paper would be well received by the community as a consequence.

I have only a few minor corrections, and suggestions.

Micropillars are compressed along perpendicular to the (100) and (111) directions. For (100), this would just be along a unit cell axis-direction, for the (111), along the body diagonal of the cube. How were the authors assured that these were the directions in the experiment? This applies to all the measurements, including fracture toughness. Notably the authors flip from (100) to [100], in this cubic structure, this would equate to the same thing, perhaps stick with [100] throughout?

Response: About the precision of the facet mounting, the alignment of the punch with the FIB samples is usually +/- 0.2 degrees. This information is now added to the Methods on micropillar compression (Page 16). In the revised manuscript, we edited the text to refer to the indented surfaces as the ‘apparent’ crystal facets of (100) and (111) respectively, as these are how the samples being mounted. Assuming a small misalignment during surface preparation (but hard to characterise this precisely), indeed the normal of (100) facet will equate the [100] axis for a cubic cell.

Page 3 In 69, how confident are you that the crystals are ‘activated’? And in what sense? Is this solvent loss as well as removal of the bound axial water molecule to reveal an open metal sit, and if so what is the evidence for this? If this could be clarified it would be very much appreciated.

Response: The activated single crystal is referred to both the solvent loss and removal of axial water molecule subject to a vacuum oven treatment under 120 °C overnight. The evidence is the reversible colour change of single crystal from light blue to deep blue (ref.48). When exposed to ambient environment, of course, the activated crystal will reabsorb water moisture, but this will desorb again under high vacuum in the FIB and during in-situ micropillar testing in the SEM.

Page 7, In 145. You state that ‘severe amorphization of the HKUST-1 structure caused by FIB milling’ is there a way to quantify this? Is anything known about amorphous HKUST-1 at all?

Response: Unfortunately there is no easy way to quantify the crystallinity of this small volume of material on the micropillar surface after FIB milling (see also response to Reviewer 2’s question 2). While the gradual structural amorphisation of HKUST-1 subject to pelleting pressure has been reported (DOI: 10.1021/acs.jpcc.9b08125), to the best of our knowledge amorphisation of HKUST-1 (or other MOFs) by Ga^+ beam has not been reported. However, MOFs are highly beam-sensitive materials, and rapid amorphisation/ transformation of a number of MOFs under e-beam is well known (e.g. DOI: 10.1002/anie.201809921, DOI: 10.1021/nn204054k, DOI: 10.1021/nl901397k).

Page 8, Figure 2. I really like this Figure, however can anything more be said, chemically about the difference observed here between the two directions? What on the surface of the crystal on these surfaces, are they different? There are three independent pore structures in HKUST-1, do you see alignment with the largest pore along the 111 giving rise to the difference? I guess what I’m asking, is whether you compress/fracture the SBU more along one direction than the other?

Response: Referring to Fig.2(a), the bond density is higher for (111) with less shear applied to the bonds and SBUs. In contrast, the bonds for (100) are oriented parallel to the shear suggesting buckling would occur, see revised Figure 2(a) illustrating this effect more clearly. In this context, the SBUs serve as nodes to facilitate the angular distortion of the parallelepiped-like structure. The precise role of the pore alignments, however, is harder to reason for a supercell structure. To gain further insights this warrants further investigations perhaps by MD simulations akin to Ref.[40]. For now, it doesn’t seem unreasonable to propose that the largest pore aligned along [111] can expedite cavity collapse and volume shrinkage in line with the H/E vs. Y/E relationship following Johnson’s theory (Fig. 2b). See additions made on Pages 8-9.

Page 14 Spotted ‘Supplementary Error! Reference source not found.’ needs

fixed. **Response:** We have corrected the error as “Supplementary Figure S1”.

REVIEWERS' COMMENTS:

Reviewer #1 (Remarks to the Author):

Authors have provided satisfactory answers to my queries and revised the manuscript. This version of the manuscript deserves publication in Communications Chemistry.

Reviewer #2 (Remarks to the Author):

The authors have addressed the reviewer comments in the revised version. I am therefore now happy to recommend it to be published in Communications Chemistry.

Reviewer #3 (Remarks to the Author):

Editorial note: this reviewer provided no further comments for the authors.